# The Case for Studying Non-Market Food Systems

**Sam Bliss** [1,2]

1   Gund Institute for Environment, University of Vermont, Burlington, VT 05401, USA; samcbliss@gmail.com
2   Rubenstein School of Environment and Natural Resources, University of Vermont, Burlington,
    VT 05405, USA

**Abstract:** Markets dominate the world's food systems. Today's food systems fail to realize the normative foundations of ecological economics: justice, sustainability, efficiency, and value pluralism. Drawing on empirical and theoretical literature from diverse intellectual traditions, I argue that markets, as an institution for governing food systems, hinder the realization of these objectives. Markets allocate food toward money, not hunger. They encourage shifting costs on others, including nonhuman nature. They rarely signal unsustainability, and in many ways cause it. They do not resemble the efficient markets of economic theory. They organize food systems according to exchange value at the expense of all other social, cultural, spiritual, moral, and environmental values. I argue that food systems can approach the objectives of ecological economics roughly to the degree that they subordinate market mechanisms to social institutions that embody those values. But such "embedding" processes, whether through creating state policy or alternative markets, face steep barriers and can only partially remedy food markets' inherent shortcomings. Thus, ecological economists should also study, promote, and theorize non-market food systems.

**Keywords:** ecological economics; markets; embeddedness; justice; sustainability; efficiency; values

## 1. Introduction

Food production uses about 40 percent of the land on earth, releases a quarter of all greenhouse gas emissions, and irrigates with nine-tenths of the world's water consumption [1–3]. It drives deforestation, toxification, eutrophication, freshwater scarcity, species extinctions, and climate change, all of which threaten humanity's collective ability to feed ourselves in the future [4–6]. Worse yet, global food systems do not even adequately feed humanity today. An estimated 821 million humans suffer from chronic food deprivation and at least a billion more experience hunger because of unequal distribution within households, high-activity livelihoods, seasonal food insecurity, micronutrient deficiencies, or intestinal parasites that inhibit absorption [7–9]. Meanwhile, 9 percent of global crop calories feed biofuel refineries and other industrial processes instead of humans. Another 36 percent feed livestock capable of digesting wild foods that humans cannot, who return fewer than one-tenth of those crop calories back to humans in the form of meat, eggs, and dairy [10]. And around one-quarter of global edible food calories, or one-third of the mass of food production, ends up wasted [11,12]. One-eighth of the world's adults are obese while more than one-fifth of children under five suffer from stunted growth as a result of undernourishment [7]. Hunger exists amid plenty, want amid waste. Food systems ransack ecosystems and fail to meet a basic human need. They are rightly already a priority for ecological economics research.

Markets govern much of global food systems. Moral philosopher Michael Sandel [13,14] calls for a public debate about where markets serve the public good and where they do not. He argues that we need moral reasoning, not just economics, to decide which social interactions and practices should be governed by market mechanisms. This debate has dealt with whether markets should govern

immigration, friendship, queues, medical treatment, university admissions, and the distribution of human organs, wombs, and blood [15–17]. Within ecological economics, lively disputes deliberate the ethics and effectiveness of using markets and market-based instruments to address environmental issues, with particular attention to monetary valuation of ecosystem services [18–21]. Jean-David and Julien-François Gerber [22] have argued that immunizing society from market dependence in general—decommodification—should be a foundation of ecological economics. Yet, whether markets for food serve the public good is a question that has been absent from ecological economics' research agenda. This article aims to spark this line of inquiry by decisively taking the negative position.

Non-market food systems, similarly, have received little systematic attention as an alternative from ecological economists. A meticulous online search using Web of Science and Google Scholar yielded just 19 articles about non-market food systems published in the journal *Ecological Economics* [23–41]. (Colleagues and I are preparing a manuscript for publication in that journal in which we enumerate the methods of this literature review in detail.) Other disciplines—anthropology in particular [42]—have studied non-market food systems, mostly in traditional societies, somewhat disconnectedly from the critique of markets. Yet people everywhere and at all times garden, hunt, fish, forage, and glean food that is not for sale. Food sharing is a universal human trait [43]. Humans share food within families more than any other mammal and between unrelated individuals in complex patterns unique among all organisms [44]. Even in the cores of neoliberal capitalism, where markets mediate most economic activity, people produce food to share, gift, and consume within the household. While the emerging body of research on urban farming and local food has focused mainly on commercial production and exchange [45–47], food systems based on reciprocity, redistribution, and self-production are nearly always local. Some of these non-market food systems may serve the public good better than markets, or as a complement to markets. Others may not. Non-market food systems deserve careful study to learn, together with their participants, about how they might promote the public good. This article, then, also makes the case for studying non-market food systems.

I begin by offering a definition of markets and reviewing some typologies of markets. Ecological economists, I speculate, have neglected to question markets for food or study non-market food systems because food is inherently rivalrous, made excludable by coercive institutions, and produced for sale in global, complex, path-dependent, power-laden systems. Yet these reasons do not suffice to make market food systems desirable; nor do they warrant omitting non-market food systems from the research agenda of ecological economics. I draw on diverse literatures to argue that markets inhibit progress toward justice, sustainability, efficiency, and pluralism—the normative ends of ecological economics—in food systems. Next, I critically examine common proposals to remedy the shortcomings of food markets: regulating or supplementing markets through the state and constructing alternative, "ethical" food markets. I argue that food systems can approach the objectives of ecological economics roughly to the degree that they subordinate market mechanisms to social institutions with other logics and values. Therefore, food systems entirely without markets are, at the very least, worthy of consideration. By way of conclusion, I propose some preliminary, imprecise outlines of a program for empirical investigation, practical action, and theory building in the realm of non-market food systems.

## 2. On Markets

A market is an institution that enables buying and selling with prices. This institution can be a physical space, a shared ritual, a set of norms, or any combination thereof. The preceding definition combines all three ways that theorists have defined what a market is [48]: based on what it looks like (prices, in this case), what it does (enables buying and selling), and what institutions or assemblages underlie it (a physical space, shared ritual, and/or set of norms). Myriad other definitions based on different observational, functional, and structural factors exist (many market theorists, however, somehow neglect to define the thing they study [48,49]). There is no "correct" definition of a market. Any delineation is valuable only insofar as it is useful for the purpose at hand. Mine will work well to distinguish market from non-market food systems because prices imply specified, generally

repeatable terms of trade. This excludes communal sharing, many centralized redistribution systems, non-simultaneous reciprocal gift exchange, and, I suspect, most barter.

There are many types of markets. Many typologies of markets, in fact. Neoclassical economics treats markets as relatively homogenous institutions. Economics textbooks tend to distinguish between types of markets based only on how they deviate from the theoretical ideal of perfect competition in a self-regulating market system generating socially optimal equilibria. Markets are "distorted" in the case of natural monopolies, common-pool resources, public goods, and the ubiquitous benefits and harms to others not involved in the transaction. Heterodox economists and other social scientists have proposed further market typologies based on completeness of contracts, the roles of participants, and other aspects that make real markets differ from market theory [50,51]. By contrast, classical political economists such as Adam Smith [52], David Ricardo [53], and Karl Marx [54] differentiated their theorizing about markets based on what was being bought and sold: goods, land, labor, and credit have different characteristics and different people buy and sell them in much different contexts and through different institutions. Echoing these thinkers' concerns, Karl Polanyi [55] argued that labor and land could never be governed entirely by markets because real people and ecosystems are not produced for sale and have needs that markets cannot meet or account for. But this arrangement is exactly what was desired by early 19th-century capitalists, who needed steady access to workers and inputs, and by economists, who urged the establishment of these markets for factors of production in order to discipline the poor and organize society in service of industry [53,56,57]. Classical economists theorized and promoted an all-encompassing self-regulating market system even as they admitted that land, labor, and money differed in important ways from other commodities. Polanyi called these fictitious commodities. He and other scholars of economic anthropology also conceptually separated external markets for trading *between* communities from internal markets for trading *within* them; societies without all-encompassing market systems rarely have the latter, internal type of markets, but instead rely on systems of reciprocity, redistribution, and self-production to meet needs and desires within communities [58,59].

Polanyi [55] saw a fundamental distinction between markets embedded in social institutions and the disembedded markets of an all-encompassing market economy that can only work properly—that is, self-regulate according to economic theory—when all other cultural and political governance mechanisms are subordinated to rules that facilitate competitive buying and selling in pursuit of gain, such that the market forces of supply and demand can determine prices and outcomes. That is, the market system requires disembedding markets from non-market social institutions. Market mechanisms then provide the set of rules guiding economic behavior and decision making. This is not to suggest that disembeddedness is the natural state of markets (if such a think were to exist). States disembed markets by forcibly creating markets for labor, land, and most everything else. This means eradicating social safety nets and establishing private property over land, among other reforms. Yet people and other beings fight back against full subjection to the whims of the market. Fully disembedded markets would destroy society and nature, Polanyi argues. So societies regulate, constrain, modify, and escape markets to retain the influence of non-market norms and values. This re-embedding counter-movement thus subordinates markets to other social institutions. One could inexactly place all markets on a spectrum between embedded and disembedded. Since markets can be thought of as patterns of behavior that follow institutionalized rules, markets embedded in non-market social institutions are extremely diverse, while increasingly disembedded markets more and more closely resemble the ultimately unrealizable theoretical ideal of the self-regulating market system. This suggests that the disembedded—really *less*-embedded—markets of a market economy represent a purer form of market, freer from the muddying influences of particular cultures, places, and non-market institutions.

Markets for foods are in many ways their own type [60]. Food is a human physiological necessity, without which we do not exist. Foods are organisms that comprise the ecosystems of which we form part. Food is the basis of rituals in every culture. Markets for other things that are essential, ecological,

or culturally important share some of the characteristics of food markets I describe below. Some aspects of food markets apply to virtually all markets. Therefore, many of the hypotheses and contentions I make below can inform a research agenda questioning whether markets serve the public good in the case of not just food but any chosen good or service, especially other essential, ecological, and culturally important resources such as housing, water, or medicines.

Food markets are also quite diverse [60]. Again, diversity corresponds with embeddedness: local, weekly (embedded) marketplaces each have their own norms and quirks, while (disembedded) markets for agricultural commodities like wheat are global and rather homogenous. But even market food exchange that is highly embedded in social relations—farmers selling raw milk at negotiated prices to their neighbors, for example—cannot necessarily be considered a spontaneous or fully autonomous phenomenon because internal markets for things like food exist only where states have turned land and labor into commodities by enclosing commons and destroying social institutions of reciprocity and redistribution [55]. Yet ecological economists have tended to treat food markets as essentially inevitable. Why have food markets hardly been questioned? How have non-market food systems escaped careful consideration?

## 3. Food Markets Seem Inevitable

Ecological economists have neglected to systematically evaluate the desirability of food markets or develop any coherent body of knowledge on non-market food systems for sets of reasons that overlap considerably. Therefore, I combine my conjectured motives for omitting these two areas of inquiry—or lacks of motives for studying them, as it were—into a single list.

First, food is a private good according to economic theory [61]. This is because food is rival and excludable. It is rival because you cannot eat food I eat. It is excludable because legal institutions backed by the threat of violence can prevent you from taking food that is my property. A systematic review of English-language academic texts since 1900 found nearly 50,000 references to food as a commodity or private good and just 179 to food as a commons or public good [62]. Authors overwhelmingly referred to "food *as*" a commodity, commons, or public good but wrote that "food *is*" a private good, suggesting that scholarly understandings of food have been dominated by neoclassical economic thinking. According to economists, private goods *should* be traded in markets. Supposedly costly non-market governance mechanisms should be "saved" for things are non-rival, non-excludable, or both. Moreover, the study of economics in general has confined itself to the study of markets [49].

Second, food is actually produced for market, unlike labor, land, money, and most ecosystem services [55]. Fellow ecological economists write, "Following Polanyi's scheme, some commodities are not fictitious; they are produced for sale and exchange. There is no problem with valuing tomatoes with money" [18]. Food itself and most factors required to produce it can be and have been private property [63]. Yet history suggests that the fictitious commodification of land and labor (which are not produced for sale) triggered the widespread development of food markets. Thus, food itself is produced for market at least in part because of the creation of markets for things that are not.

Third, ecological economists may believe that markets are the least bad of all options for governing food systems. Food systems are complex, and markets simplify exchange and obviate continual deliberations. Coordinating production and distribution across space and time is difficult without markets [18]. Markets have existed before and outside of capitalism [55]. This all contributes to the seemingly pervasive belief that markets and central planning are the only two options for coordinating economic activity in large-scale societies [64]. Maybe scholars are unable to imagine widespread, desirable, non-market, non-state food systems, and thus discount existing examples as uninteresting or not useful for thinking about systemic alternatives. Or maybe they can envision such systems but consider them unrealistic or bound to fail.

Fourth, the omission may be pragmatic. The hunger and ecological destruction in the food system are urgent and economic institutions exhibit considerable path dependence [65]. Authors may feel compelled to propose remedies that can be implemented in today's capitalist world. More to the

point, researchers may be trying to come up with solutions that are attractive to actors in positions to enact sweeping changes; they may be pandering to people in power with politically palatable reforms. Ecological economists tend to carry out research that is relevant to designing government policies and programs. Findings about non-market food systems will often seem relevant only to their participants.

Finally, the beneficiaries and proponents of market food systems have political power in the academy and in society. Some parts of this phenomenon feel rather innocent. Market exchange of food eludes examination perhaps because buying and selling food feels natural; most people trade money for food in the marketplace almost daily in the urban areas where universities are located. Similarly, non-market food systems might be cast as hobbies or marginal sources of nutrition because this is how prominent scholars experience them. Research tends to reflect the worldviews researchers have been trained to accept and adopt, both in their academic formation and through life experience [66]. Ignoring non-market food systems, for example, reinforces the marginalization of unpaid, reproductive work in capitalism. I explain below how market food systems serve and reinforce the dominant systems of power, which play an outsize role in determining research agendas—and, indirectly, findings—through state and philanthropic disbursement of funds as well as via the institutions that assign academic prestige. Social facts cannot be separated from values [67], and interrogating market food systems or exploring non-market ones may well uncover facts that threaten the values of those in power.

To sum up, food is a rival good that political institutions make excludable. It is produced intentionally for sale in markets that would be quite difficult to abolish or replace. These are necessary but not sufficient criteria for justifying the ecological–economic desirability of market food systems. They in no way refute the call to systematically analyze, encourage, and theorize non-market food systems. Indeed, the role of uneven power relations in setting research agendas suggests that ecological economics' purported normative orientation toward justice might by itself motivate studying food systems without markets.

Table 1 proposes a rubric for determining the goods, services, and resources for which markets can serve the public good. This scheme loosely draws on the criteria for deciding whether to accept monetary valuations of resources or ecosystem services proposed by Kallis and colleagues [18]. Ecological economists to date have concentrated on the first four criteria—rivalry, excludability, non-fictitiousness, and complexity. They have also frequently supposed, often without stating so, that the corruptness criterion is met, by assuming that governments are the only available non-market institution for managing economic systems and that they would generally do so less desirably than decentralized market coordination. The desirability criterion is novel; it adapts Sandel's conception of serving the public good to the normative foundations of ecological economics. In the remainder of this article, I will argue first that disembedded food markets do not meet this desirability criterion, and then that ecological economists should study non-market food systems to reflect on when and where they can, or do, perform more desirably than market food systems.

**Table 1.** Criteria for assessing whether markets serve the public good.

| Criterion | Markets Should Govern the Production and Exchange of Something if . . . |
|---|---|
| Rivalry | its consumption subtracts from it or prevents others from consuming it |
| Excludability | institutions can prevent specific actors from accessing or consuming it |
| Non-fictitiousness | it is produced for sale |
| Complexity | it is produced and exchanged in complex networks of actors |
| Desirability | markets can promote justice, sustainability, efficiency, and value pluralism |
| Corruptness | non-market institutions would do so undesirably |

## 4. Methods

In what follows, I assess the desirability of market-based food systems according to a set of objectives derived from Daly and Farley's [68] three goals for ecological economics—justice, sustainability, and efficiency—plus another foundation of the discipline: the weak comparability of

values [67,69]. For the latter, I examine markets' effect on the plurality of values in food systems; since values are incommensurable, governance must take each into account separately. I occasionally use the term "desirable" in this article as a shorthand for just, sustainable, efficient, and value-plural. In the spirit of pluralism, I do not contain my arguments to single definitions of justice, sustainability, efficiency, or pluralism itself, but instead explore how market mechanisms interact with multiple conceptions of each of these foundations.

My argument is decidedly one-sided. The reader interested in reviewing the theoretical advantages and well-rehearsed defenses of markets should consult any elementary economics textbook, or marketing materials from food-related firms like grocery stores or packaged snack manufacturers. My purpose is to synthesize theory and evidence from diverse disciplines to call food markets' desirability into question. Some of the arguments to come refer to characteristics and consequences specific to the disembedded markets of a market system. If we suppose that markets have some generalizable properties, functions, or at least regularities beyond the content of the definition I have proposed, then it follows that markets would exhibit these characteristics in proportion to the extent that a society subordinates other institutions to the rules of markets. The disembeddedness of markets vaguely corresponds to the "marketness" of food systems, or of society [70]. We see evidence for the generalizability of markets in the fact that capitalist markets in the neoliberal era, when seemingly everything is for sale, display far less variability than those of traditional societies, where markets are limited to special purposes and are subject to strict norms regarding what can be traded, when, where, how, between whom, how much, and at what terms of trade. Thus, this assessment refers mainly to the generalities of the former.

However, since no market fully realizes the impossible ideal of disembeddedness, my contentions capture *tendencies in the disembedding process.* That is, as non-market social institutions are increasingly subordinated to market rules, those markets increasingly resemble the markets I describe in the next four sections (on justice, sustainability, efficiency, and value pluralism). Because neoclassical economic theory portrays imaginary, perfectly disembedded markets, my assessment draws several insights from it. Yet I rely more on the critical conceptions of markets from heterodox schools of thought and other social sciences. In actually existing market economies, unlike in theory, markets have money and they include not just producers and consumers but participants whose sole aim is to increase their initial stock of money through buying and selling—merchants, capitalists, and speculators, namely [71]. Because markets are political and cultural institutions involving interactions between real human beings, I incorporate understandings of markets from political economy, history, anthropology, evolutionary biology, psychology, sociology, and behavioral sciences. Market food systems, like all food systems, necessarily involve human–ecosystem interactions; so the following sections also employ findings from agroecology, earth systems science, political ecology, conservation biology, and other environmental sciences.

My approach is one of critical realism. This meta-theoretical position in the social sciences calls for identifying and illuminating the structures and mechanisms—those of markets, in this case—that play a part in producing phenomena of interest, such as the injustice and unsustainability of the world's food systems [72]. The responsibility of markets for creating, or at least facilitating the creation of, the undesirable state of today's food systems has been severely underestimated, or at a minimum underexplored. As a preanalytic vision, critical realism posits an objective reality that humans can know, but never with full accuracy or certainty [67]. This assessment of food markets is transdisciplinary because molecules, cells, brains, organisms, societies, ecosystems, earth systems, and so on are ontologically different, and so must be studied by a plurality of sciences. Moreover, different, at times incompatible, ways of knowing, approaches to inquiry, and even beliefs about reality can each be useful for forming tentatively reliable understandings of diverse facets of a world that is ultimately unknowable. They can challenge or substantiate each other's truth claims, or create new knowledge together. The research agenda I propose thus encompasses plural philosophical perspectives. Multiple

methodologies are needed to study if, when, where, why, and how market and non-market food systems serve the public good.

## 5. Results

### 5.1. Food Markets Are Unjust

Markets are procedurally unjust because they give actors say over economic decision making in proportion to their purchasing power and access to capital for investment. This allots power to the wealthy. Markets warp food systems, and entire economies, toward what rich people want. They are political institutions. Where economic inequality exists, markets are undemocratic since they operate on the principle of one dollar, one vote. They remove collectively important choices from the realm of public deliberation and decision making, handing the reins directly to property owners [73]. Additionally, market power, incomplete contracts, non-clearing markets, and other conditions can make one party to an exchange dominant over another [50]. These power imbalances can undermine the voluntariness of exchange [14]. In the case of food, exchange is rarely fully voluntary: one cannot choose to refrain from eating, and non-market options are often limited. Food producers cannot choose to refrain from selling, either. They must accept market prices in exchange for their produce to maintain their livelihoods and pay for inputs—land, labor, water, chemicals, and seeds, in the case of farming. In economic terms, the bargaining power of parties with perfectly inelastic demand or supply is functionally eliminated. Involuntary exchange is ripe for exploitation.

Markets thus create distributive injustice, too. They channel benefits to actors in proportion to their purchasing power, which does not accurately reflect their needs, their equal share, or even their contributions to society (these correspond to the three most typical principles of distributive social justice: need, equality, and equity [74]). Markets do not distinguish between luxury and sufficiency; food goes to whoever can and will pay the market price. This systematically punishes markets' poorest and most marginalized participants. Many people cannot afford enough market food to meet their basic nutritional needs. Meanwhile, others pay to overeat, waste food, and direct edible crops to livestock and biofuel production [63]. Not just the quantity but the quality of food is distributed unjustly: the world's urban poor tend to eat addictive, unhealthful, ultra-processed food-like products manufactured from cheap cash crops [75]

Markets exacerbate economic inequality over time, making distribution increasingly unjust. Economies of scale favor bigger farms, distributors, retailers, and input producers over smaller ones. These big players consolidate their power through a self-reinforcing feedback loop: large retailers prefer to source from large wholesalers, who buy from large processing firms, who contract with large commodity traders, who buy from large industrial farms, who get their inputs from large transnational chemical corporations. This simplifies administration and decreases the costs of regulatory compliance. It also puts up barriers to market entry for smaller enterprises, who must compete with oligopolies and oligopsonies of transnational corporations. Independent producers get squeezed at both ends by powerful megafirms, constraining smallholders' revenue and farmworkers' wages [63]. Power dynamics in bargaining favor wealthier actors. Farmworkers, for example, risk being fired when they demand decent pay and humane working conditions. Historically, income inequalities have risen sharply when protections have been removed from market economies, such as during the neoliberal era of the last four decades [76,77]. Inequalities within countries are at levels not reached since the early 1930s, and inequalities between countries remain high despite, or perhaps because of, the globalization of markets [78]. Finally, market settings might make people more comfortable with distributive injustice than they otherwise would be. Across cultures, framing economic experiments as markets leads participants to quickly converge on highly unequal equilibria [79].

Market settings promote behavior that produces injustice by forcing people to try to maximize what they get and minimize what they give [80]. This encourages an antagonistic ethic. Merely prompting people to think about money makes them offer less help to others, ask for less help from

others, and be generally less cooperative, caring, and warm in experiments [81,82]. In market settings, people tend to reallocate their time and effort from relational investments like trust and community to general investments like their own education [83]. Markets can even trigger moral decay. Some research suggests that markets make people act more selfishly [84]. Market settings seem to enable people to justify actions that in other settings would be unjustifiable [85,86]. The pressures of competition, in particular, can bring about a proliferation of unethical behavior [87]. In an economic experiment involving lab rats, auction markets made people significantly more willing to let a rat die for a given sum of money compared to an individual, non-market condition [88]. In the real world, nearly everyone purchases market food from industrial systems that brutalize domesticated animals, drench ecosystems in poison, and undercompensate vulnerable humans working in often appalling conditions. None of this should be surprising. In a market that resembles that of economics textbooks, such as buying fruit from one of many vendors, actors are in a psychological environment characterized by anonymity, self-regard, mobility, independence, isolation, and calculation [50]. Some sociologists suggest, after all, that markets are to some extent a performance of economic theory [89]. Markets affect behavior beyond the marketplace, too, because they are cultural institutions. Our activity in markets contributes greatly to making us who we are. Markets create people [90].

Markets create people who are more likely to tolerate and generate injustice. Markets reduce local material interdependence, social solidarity, and practices of generosity, since buyers and sellers need not know or care about each other or remember previous transactions [90–92]. To the extent that markets' extrinsic rewards affect people's motives to act responsibly, they probably crowd out intrinsic motivations [93–95]. Because the tasks we perform influence the people we become, the fact that markets ask so little of us ethically suggests a reduction in both the salience of moral concerns and the capacity for moral reasoning [50,83]. Indeed, experiments have found that institutions that align individual and collective incentives—as do markets, in theory—create a barrier to learning altruistic behavior and moral reasoning [96]. It is not necessarily a good thing to economize on solidarity, empathy, communication, generosity, and collective decision making. These are not scarce, rivalrous resources but muscles to be trained [14]. Markets contribute to their weakening. "If friends make gifts, gifts make friends," wrote Marshall Sahlins [58] of his experiences in tribal society. If strangers make market exchanges, then market exchanges make strangers, added Sam Bowles [50].

Markets might make societies more unjust for evolutionary reasons as well. Natural selection has acted on human evolution at the group level primarily, including by promoting cooperative cultural adaptations [97]. Human social groups transmit culture via sets of norms regarding what is and is not acceptable behavior [98]. Economic institutions can affect cultural evolution in two ways: rewards and conformism. In theory, the presence of market institutions leads to a lower equilibrium population frequency of cooperative, prosocial traits [83]. The more marketness an economy exhibits, the more strongly forces of natural and cultural selection select for individuals who are self-interested or competitive. All else equal, increasing marketness also reduces the frequency of repeated pair-wise interactions, which makes it more difficult to sanction violations of norms and reward good behaviors with reputation. Moral economies that exist to make sure no one goes hungry unless everyone does eventually break down or wither away [55,99]. If market elites emerge, they use their status to entrench and expand markets [50,64]. If society must protect itself from the harmful consequences of markets, then these counter-movements, to the extent that they are successful, further entrench markets by making their effects more acceptable [55,100]. Practices that promote injustice root themselves into the social fabric.

Markets legitimate unjust social relations, too. Across cultures, mythologies surrounding value tend to conceal the collaborative nature of its creation [90]. Markets and market-centric economic theory devalue and invisibilize the unwaged labor of women and non-humans that supports all production [101–104]. Markets degrade the unpaid work that sustains them. It is market economies' magic that they can pretend to be about something other than making people and social relations [50,59]. Moreover, markets make inequality and exploitation appear as spontaneous results of countless

voluntary economic interactions, rather than as the outcome of any organized decision-making process. Thus, injustice can feel justified or inevitable. This phenomenon might explain the finding that people are more willing to consent to injustice in market environments. Producers in food markets, for example, struggle to provide both adequate wages for farmworkers and affordable food for low-income families. These two objectives contradict each other. To farmers, this conundrum could feel like an iron law of agriculture rather than an attribute of a specific economic institution.

Markets create further distributional injustice when accounting for nonparticipants. Markets facilitate and reward imposing costs on third parties [105,106]. Competition drives market actors to shift burdens onto the public [107]. This process most affects those without the political or economic power to prevent others from taking resources from or dumping wastes on their environments [108]. The world's poorest people are the most dependent on the ecosystems that the global food system pollutes and degrades [109]. They are also the least able to afford artificial substitutes to ecosystem services, to the extent that those exist [21]. And, as suggested above, market participation might make people care less about resultant injustice, because markets diffuse responsibility [110] and evoke selfish, materialistic values. The ways that markets enable, normalize, and entrench injustice mutually reinforce one another.

*5.2. Food Markets Are Unsustainable*

Markets enable and reward environmentally harmful practices. Market food producers must make their operations financially viable to exist. They have to produce at lower cost than the market price. Likewise, markets direct agronomic research, breeding programs, and technological innovation toward money-making rather than alternative objectives. These incentives for cost-cutting in fact encourage high-input production methods because machinery and chemicals are cheaper than labor and land. A barrel of oil can do the work of 20,000 hours of human labor. Fertilizer can replace leaving fields fallow for fertility. Society bears many of the costs imposed by machinery, fuel, pesticides, antibiotics, fertilizers, and irrigation systems, in the form of pollution, resource depletion, and the degradation of ecosystems on which humanity depends. As more producers adopt cost-cutting—more accurately cost-shifting—practices, competition drives down food prices and then all must adopt these practices. If many succeed in increasing production, the oversupply pushes prices down as well [63]. Farmers, fishers, and foragers must produce ever more cheaply to stay afloat, forever prioritizing short-term financial viability over long-run environmental sustainability and other goals [111]. Over time, market mechanisms have facilitated the industrialization of the food system, which has wrought havoc on ecosystems around the world.

Markets facilitate the surpassing of sustainable scale, moreover. By continually motivating and delivering productivity increases, competitive markets free up labor and resources to produce an ever-greater array and quantity of goods and services. Wave upon wave of workers flock to cities as food production becomes more labor-efficient, constructing industry and services on top of an economy's agricultural foundation. Increases in material efficiency beget growth that ultimately overwhelms those efficiency gains, increasing overall resource use [112–114]. Likewise, yield increases have not spared land from agricultural encroachment [115,116]. Deforestation in fact tends to increase as production per hectare increases or rural population decreases [117,118]. That productivity gains make farming temporarily more profitable leads to more land in production [119]. Then, as prices fall in response to oversupply, farmers need to increase production further to make enough income to support their livelihoods [63]. At the macro level, increasing land and resource productivity backfires—that is, leads to greater overall land and resource use—because these efficiency gains drive growth. Cheaper food leaves consumers more money to spend on everything else.

Markets also enable economic growth by making exchange itself more efficient; they decrease transaction costs relative to sharing and gift exchange, which require established social relationships, some degree of trust, and often cumbersome rituals. State-instituted markets enable trade between strangers and over distance, allowing for increasing specialization and consequently greater economic

efficiency. All these efficiency gains make food cheaper and more abundant, increasing real incomes and human population, the twin components of growth. Cheap food and raw materials drive profits as well, increasing the expansion of capital that propels growth in the long run, pushing human environmental stressors past biophysical thresholds of sustainability [120,121]. Some might protest that these are effects of capitalism, not markets as such. Yet, in the sphere of agriculture, the two are inextricably intertwined. World-ecological theory traces the origins of capitalism to market-oriented plantation agriculture, whose slaves were in some instances forced to live by purchasing market food with wages [122]. Similarly, political Marxist theorists argue that capitalism first arose from the genesis of market dependence [123] and market imperatives [124] in the livelihoods of peasants. Once households rely on market exchange for their social reproduction, the pressures of external competition force them to "improve" the production process systematically and continually. In sum, the actions that food system actors, individually and collectively, must take to survive and succeed in market settings align with neither local environmental protection nor planetary sustainability.

Markets, moreover, fail to punish the surpassing of sustainable scale. They rarely signal scarcity or degradation of ecosystems that freely provide resources and services essential to humanity's survival. To be sure, policy can create artificial market mechanisms that govern the use and maintenance of ecological systems whose benefits to humanity are not tradable private property. Yet treating ecosystem structures and functions as market goods is a risky abstraction because ecosystems provide a complex multiplicity of interrelated attributes that benefit people in critical ways we often cannot comprehend until they fail [55,106,125,126]. At the very least, economic actors will respond to market-based environmental policy by perpetually shifting costs elsewhere, onto nature or people that have little or no market value. A literally all-encompassing market system—one in which *all* of nature's benefits and costs to people are bought and sold, or artificially priced—would greatly extend the injustices of markets that I described above: individuals' environmental preferences would count in relation to their purchasing power and poor people would be forced to cut consumption to reduce their already-meager environmental impact while the rich simply pay to pollute [21]. Such an arrangement, internalizing every externality, is not remotely possible, anyway, since many of nature's values are neither compatible with property rights nor straightforwardly quantifiable, much less commensurable with dollar values [67,69]. Monetary valuation of environmental goods and services can never capture all of the relevant information for decision making [106]. Some scholars, furthermore, argue that the market system would fail if capitalists were to have to pay the full social costs of production [120]. The earth's life-support systems will likely fail first, perhaps irreversibly, if the environment is protected in relation to its imputed market value, since most individual components of ecosystems are valuable in their functional relations to the whole, not because of their specialized, separable properties, much less any attributes that individual consumers can enjoy.

Markets, by promoting specialization, make food systems more vulnerable to the environmental disturbances they contribute to causing. Commercial production of cash crops and standardized livestock have displaced and eliminated countless crop varieties and animal breeds that provided stable subsistence but not profits through sale [127–130]. The loss of agrobiodiversity, including genetic diversity within varieties, robs humanity of genetic resources from which to breed new foods fit for a changing global environment [131–134]. Specialization of land use and labor means monoculture and mechanization, which reduce local biodiversity and preclude labor-intensive agroecological farming techniques. Since densely populated single-species landscapes tend to host more pests and diseases than biodiverse ones [135,136], pesticide use becomes indispensable. This speeds up pest evolution: resistant insects, weeds, and bacteria typically appear within a decade of new insecticides, herbicides, or antibiotics, sometimes sooner [137]. Superweeds and superbugs threaten to destabilize an increasingly homogenous global food system. And specialization has produced an increasingly urbanized human population that fundamentally depends on large-scale, highly productive agriculture.

Market food systems have produced a particularly confounding sustainability and resilience predicament related to nutrients. Urbanization breaks the nutrient cycles in which humans participate.

Nitrogen and phosphorous leave fields via harvests and leaching, never to return. Potentially valuable nutrients become unwanted waste in cities and on industrial farms. They create massive dead zones in aquatic and marine ecosystems. Farmers must continually apply fertilizer, mostly from non-renewable sources. At this point, synthetic fertilizer manufacturing alone produces almost twice the estimated sustainable limit for adding nitrogen to the global environment [4,138]. Yet suddenly subsiding fertilizer production could condemn hundreds of millions to starvation, since synthetic sources now account for about half of the nitrogen in the proteins that make up human bodies, according to one estimate [139]. These results of specialization are historically contingent and cannot be attributed to markets exclusively, yet it is clear that the imperatives of a market economy and food system—increase productivity and decrease the proportion of the workforce dedicated to feeding the population—restrict the option space for addressing the breakdown of nutrient cycles.

Market food systems might make societies less resilient overall. It is clear that extreme specialization will hinder societies' ability to recover from, and react to, the global environmental changes and concomitant civilizational catastrophes that earth systems scientists foresee [140,141]. Peer-reviewed comparative case studies show that diverse, ecologically complex farming systems sustain less damage than simplified ones in extreme weather events like hurricanes [142,143]. Labor specialization has left much of humanity deficient in the sorts of food-related skills and ecological knowledge that may be necessary to survive and thrive in a changed climate [144]. Market-mediated economies deprive communities of the intimate social relations and spirit of mutual aid that can ensure collective food security in times of crisis or shortage [145].

What is more, markets actively obstruct society from effectively addressing environmental problems. Markets cut off exchange of all information between production landscapes and consumption centers other than commodities' price, quantity, and observable characteristics [146]. City dwellers purchase food whose production may be invisibly unsustainable [147]. Yet demand from consumers and profit-oriented food manufacturers guides production decisions more than the intimate ecological knowledge of farmers, fishers, and foragers. Cognitive, institutional, and ethical lags separate initial, proximate benefits from eventual, distant costs, breaking feedback mechanisms between production, distribution, and consumption [148]. Placing responsibility for, and addressing, unseen damage wrought by many hands in service of markets poses substantial challenges [149]. Plus, markets impede the cooperative attitudes and behaviors necessary to address humanity's sustainability challenges. Only international cooperation can solve global public goods problems like climate change and, to some extent, hunger [150]. Yet individuals and nation-states stand to benefit by acting in their own self-interest. These are prisoner's dilemmas [21]. To address such issues, society must create economic institutions that promote cooperation and altruism, not antisocial behavior [151].

Finally, markets for food might inhibit the formation of environmental values. Interacting with the living world and experiencing the negative effects of environmental degradation tend to correlate with pro-environmental attitudes, worldviews, and behaviors [152–155]. Markets, however, distance consumers from both the ecology of food production and the environmental damage it causes [146]. Those with the most purchasing power—the actors whose preferences essentially *design* market food systems—will also tend to be most protected from the environmental consequences of their decisions, which presents not only a barrier to developing pro-environmental values but also a major problem of moral hazard. Some studies suggest that egoistic motivations, which markets promote, reduce environmental concerns [156]. Thus, market mechanisms in food systems not only contribute to creating environmental problems and impede societies from resolving or dealing with them; they make it easier for people not to care about the environmental consequences of their food.

*5.3. Food Markets Are Inefficient*

Markets for food are inefficient because price signals frequently fail to elicit allocation responses as theorized by welfare economics. During times of shortage, a rapid price rise does not necessarily constrain excessive consumption. In the short run, food demand is inelastic because it is largely

determined by habits, culture, and necessity. Rich people do not cut consumption much in response to price escalations because food expenditures comprise a tiny share of their incomes [21,157–159]. Thus, when food prices increase suddenly in response to, say, a failed harvest, limited supplies of staple grains end up allocated to large livestock and processing companies while food-insecure households are forced to purchase less for lack of money. Price spikes cannot easily spur increased food production, either. It takes at least a growing season for producers to increase the quantity of crops supplied to market. Moreover, individual farmers do not always produce more in response to higher prices, since their incomes increase just by producing the same amount; peasants, in particular, exhibit such satisficing behavior when they can support a decent livelihood with less work [160]. And barriers to entry, notably access to land, prevent new farmers from quickly ramping up production.

Markets for food are unstable as a result. Since supply and demand do not quickly adjust, small disruptions to food production can cause wild price escalations. Speculators purchase food when prices begin to rise, knowing they will keep rising. This further increases food prices, in turn begetting more speculative demand [161,162]. Even in local markets, merchants and farmers can hoard food when it begins to seem scarce, exacerbating shortages into crises [163]. Price spikes make the poor not just hungrier but poorer, in terms of real income. This holds for poor farmers, who are often net buyers of food [164]. Putting all the food access eggs in the market basket, so to speak, may make society more vulnerable to food price shocks, which are becoming more frequent in an increasingly changed climate [165]. Markets' instability undermines their ability to efficiently guide resource allocation toward food production and distribution.

Market efficiency evaporates entirely if we allow minimal interpersonal comparison of utility. At the margin, markets tend to allocate essential, non-substitutable resources like food to those who least need them [21]. As a person nears starvation, food's contribution to their well-being becomes immeasurably large [166,167]. Yet markets send the marginal unit of food to well-fed, or overfed, people for whom its value is miniscule if not negative. This happens because undernourished people tend to have little purchasing power [168]. If they had sufficient money, they would be able to buy sufficient food. Thus, reallocating food from overnourished to hungry people will increase total utility (the law of diminishing marginal returns make this obvious). Pareto forbade comparing subjective satisfaction of subjective desires but prioritizing physiological needs over psychological preferences feels ethically defensible, if not imperative.

Markets do not even efficiently satisfy any set of predetermined preferences weighted by purchasing power, because they are cultural institutions. Markets shape people's preferences [83]. Preferences become increasingly endogenous in relation to the marketness of society. "It is uncomfortably circular to justify a set of market arrangements on the grounds that they promote the satisfaction of preferences if those preferences are themselves substantially the result of the very market arrangements under question," writes Sam Bowles [50]. Ecological economists have made the same point [68].

Markets are not entropically efficient, either, to the extent that they motivate substituting energy-intensive inputs and machinery for human and ecosystem work. Modern, market-oriented food production systems turn energy inputs into edible calories much less efficiently than traditional and subsistence farming methods [169–171]. Massive amounts of food end up wasted because actors receive no reward for ensuring that edible-but-not-sellable food ends up feeding people. If we redefine food system efficiency with human nourishment in the numerator and ecosystem services sacrificed in the denominator, then market allocation fails spectacularly. Given that the pursuit of certain agronomic efficiencies has to some extent enabled the explosion of food waste, the escalation of obesity, and the surpassing of sustainable scale, the ambiguous goal of efficiency itself should be questioned [172]. Really, food system actors continuously balance the pursuit of different efficiencies and other values, some of which contradict each other.

### 5.4. Food Markets Suppress Value Pluralism

Markets organize food systems according to exchange value, neglecting food systems' cultural, spiritual, physiological, and ecological aspects. Values are the things people consider important. Food systems, like the environment writ large, are sites where conflicting values and interests compete [107]. The communities and stakeholders that hold competing values also compete: conservation organizations might prioritize biodiversity and recreation, while farmers might struggle for material ecosystem services and the aesthetics of a working landscape. Decisions, then, distribute different goods and bads across groups, through space, and over time [173]. Above, I made the case that markets give disproportionate decision-making power—and thus disproportionate benefits—to the wealthy. But even in a hypothetical scenario with perfect economic equality, markets systematically distort decision making toward certain values, undermining others. This is because markets value things in monetary terms, when in reality values are weakly comparable and therefore not commensurable via any single, common metric [69].

Markets seek to commensurate incommensurable values into prices, denominated in dollars or any other currency. Imagine a food producer is choosing whether to plant an apple orchard or a rotation of cereal crops on a plot of land. Cereal crops would produce more food in the first five years while the apple trees mature, yet the orchard would produce far more calories over a 40-year period. But cereals are easy to store and transport; they require less water and fewer pesticides; and they provide more protein, complex carbohydrates, essential minerals, and B vitamins than apples. On the other hand, apples are a better source of vitamin C, vitamin K, and potassium. And orchards can host more plant diversity and better bird habitat than fields of grain. They need not be tilled, but still typically require more human labor than cereal crops, which are more easily mechanized. Each option performs better on some values and worse on others; many are context-specific. Some cultures might ascribe beauty to amber waves of grain, while others might organize festivals around harvesting apples and pressing cider. There is also an aspect of uncertainty: the apple trees, for example, could be invaded by pests and die before producing at all. Clearly, there is no "right" decision or objective basis on which to decide. No solution optimizes all criteria at once. A compromise solution depends on the values that food producers, and society, place on these different attributes. These diverse values lack a common unit of measurement—material, ecological, energetic, or otherwise [69]. In a market food system, these sorts of decisions are made primarily based on profitability; monetary exchange value is used as the de facto common denominator, even though people are neither cognitively nor ethically comfortable with transforming a complex of relations into the single metric of money [106]. Value pluralism, by contrast, seeks to understand the diverse ways that humans give importance to things, recognizing that these values often conflict, are incommensurable, and cannot be reduced to any ultimate value [155].

Markets inhibit justice, sustainability, and efficiency in part by suppressing the values important to achieving these goals. Even though different values and aspects of food systems are not commensurable, there can be hierarchies of values [69] (this article, for example, prioritizes the values of justice, sustainability, efficiency, and pluralism). One might consider crop production for meeting human nutritional needs a higher, more important value than for making automobile fuels. Yet markets frequently prioritize the latter; unlike feeding the poor, it is lucrative to sell crops to powerful refining corporations set on meeting a demand for biofuels backed by the purchasing power of the vehicle-owning American middle class. Market farmers must care for plants and animals according to what is profitable or feasible in terms of monetary *value*, not what is desirable according to incommensurable *values* like religious beliefs, political ideologies, aesthetic preferences, or personal morals [59,174,175]. Markets for food can only operate where participants regard certain aspects of reality such as crops, livestock, water, land, and even time as commodities rather than as sacred entities or kin [106].

Markets thus turn living beings and labor into things, useful toward the self-interested pursuit of gain yet alienated from their social and ecological relationships. This promotion of *instrumental values*,

as described in the sections above, corrupts non-market norms, motives, and principles worth caring about: love, duty, care, peace, reciprocity, mutual obligation, informal exchange, and so on [13,14,83,93]. These are examples of *relational values*, which concern the relationships and responsibilities that connect people to one another and to non-human nature [176]. These values include the fundamental conditions of existence and cultural conceptions of the good life. Some scholars categorize instrumental values within relational values, since they too emerge from relationships—i.e., between subject and instrumental object [177]. (To philosopher Samuel Alexander, all values "arise through the combination of mind with its object" [178].) But it is the set of non-instrumental relational values (hereafter simply relational values) that reflect the intuitive ways in which most people understand the world, make decisions, and tell right from wrong [179]. These values, despite being ignored and repressed by market logic, are held by diverse people around the world and do motivate action to protect ecosystems [155,179–182]. Many worldviews root their cultural identities, notions of the good life, and well-being in relationships. Prominent scholars hold that nature's relational values underlie environmentalism, and that the heavy focus on conserving biodiversity (nature's intrinsic value) and ecosystem services (nature's instrumental value) is eroding the movement [176]. Others argue that relational values are the only ones fit for an environmental ethic and aesthetic that addresses the twenty-first century's crises [177]. Relational values integrate and invigorate intrinsic and instrumental values: it is the orchardist's relationship to the orchard that makes it both sacred and satisfying to her.

Markets discount relational values because they each correspond to different languages of valuation. Relational and instrumental values coexist in economic systems [183]. But markets, as value-articulating institutions, capture exchange value, not the fundamental interdependence that constitutes everyone or the eudemonistic relationships that constitute the good life. Markets reward food producers and distributors for fulfilling others' instrumental values, which incorporate relational values only to the indirect, limited extent that people's purchasing preferences reflect them. Even if some policy mechanism endeavored to assign all relational values a monetary worth and include these in market prices, the reflection would remain partial because markets change the character and meaning of relations, in part by making all things substitutable. Market-based frameworks for protecting the environment reduce complex relational values to their subset of instrumental values, which treat ecosystems and food systems as simply means to meet human preferences, interchangeable with other means toward that end [177]. It is doubtful that any market value could pretend to approximate the intangible, unquantifiable values through which food systems promote real well-being: connectedness, community, cultural identity, sense of place, and other psychological relationships. Not without corrupting or instrumentalizing them, at least.

Markets even seem to degrade relational values over time. Food markets replace producing and sharing non-market food, practices that connect people to each other and to ecosystems. The broad relational value of subsistence from an ecosystem transforms into the purely instrumental value of sustenance from a store, interchangeable with similar food from anywhere, produced in any way. Connection to place and local uniqueness have been lost as crops, livestock breeds, recipes, even microbes have been standardized for instrumental reasons. Market food systems threaten to strip the social and cultural significance from eating, degrading it to mere feeding [144]. Organic foods provide an example of markets reducing relational values to instrumental ones. Organic certification schemes and labels were originally conceived to create a separate market for capturing the value of farming practices that nourish the soil, care for non-human beings, and enact other relational values between the land, farmers, and urban consumers. Yet now organic food is overwhelmingly marketed as a way to protect the consumer's body from harmful agri-chemicals and supposedly dangerous genetically engineered crops, reflecting purely instrumental values. When people believe they are purchasing a personal protection against environmental danger, they become less motivated to act to protect the environment or address its destruction [184].

Markets might not just amplify but also breed instrumental environmental values that see nature as nothing but a useful stock of resources, sinks, services, beauty, and recreation opportunities [182]. I

argued above that markets constrain the emergence of environmental values. Market food systems at their most disembedded prevent consumers from witnessing and participating in the transformation of living beings into food. By disconnecting eaters from the landscapes, ecosystems, and farmworkers that produce their nourishment, markets hinder the development of relational values that underlie the continual struggles of communities to preserve the conditions of common existence and, if possible, pursue the good life together. Values influence decisions and behavior [185], which in turn determine the justice, sustainability, and efficiency of food system and ecosystem outcomes.

## 6. Food Markets Are Difficult to Fix

As markets approach the disembedded ideal of economic theory and neoliberal practice, they also tend to approach the unjust, unsustainable, inefficient, instrumentalist archetype described in the previous subsections. I have shown that proposals to remedy some of these problems can worsen others. Incorporating ecological costs into prices to improve sustainability, for example, reinforces instrumental environmental values [182] and intensifies the injustices of markets [21]. Increasing incomes until everyone can afford sufficient market food, in the name of justice, would in turn accelerate the surpassing of planetary sustainability boundaries [186]. Ecolabels and alternative "ethical" markets—organic, fair-trade, and the like—seek to value plural values like justice and sustainability, typically through price premiums, yet in so doing restrict virtuous choices to affluent people seeking green status [187]. At worst, they enable consumers to reproduce unjust social relations while believing that they are undermining them [188].

Yet some of the justice- and efficiency-related problems of markets could be unambiguously assuaged by radically reducing economic inequality through the redistribution of existing income and wealth, including land. As a society approaches perfect wealth and income equality, "one dollar, one vote" comes to resemble an equitable economic democracy. More-equal societies outperform less-equal ones on all sorts of indicators of social, psychological, and physiological health [189]. But reducing inequality is unlikely. Stanford historian Walter Scheidel [190] finds that established inequalities have been flattened in the past only by mass-mobilization wars, transformative revolutions, state collapse, and catastrophic epidemics. French economist Thomas Piketty [77] has partially replicated these findings in wealthy countries over the past several centuries. He also showed, as mentioned, that market economies tend to exacerbate inequality over time. Even where better-intentioned states have attempted to redistribute land and enact other progressive reforms, more-powerful foreign interests have often forcibly imposed capitalist development, providing ideological justifications for intervention through departments of economics in universities and government. The MIT Center for International Studies in 1957 proposed "deeper military involvement in rural development so that peasants would be less inclined to support 'internal insurrections'" [163]. Those whom inequality favors control the distribution of wealth. They did not ascend to their elite positions through generosity. To be sure, reducing inequality is a worthwhile perennial effort not just as a means to make food systems and markets more desirable but for its own sake—that is, to achieve distributive justice and egalitarian societies. I leave it aside here as a separate struggle that is on its own insufficient to resolve the undesirable qualities of market food systems described above. Note, however, that subordinating markets to egalitarian social institutions can make societies more equal even in the absence of income or wealth redistribution. In societies whose markets are more embedded in institutions that treat individuals as equals, a given level of *economic* inequality will correspond to less *social* inequality.

Societies can make their food systems generally more desirable by embedding markets in desirable social institutions. My argument, in other words, is that societies limit the injustice, unsustainability, inefficiency, and value monism their food markets perpetrate and facilitate by intentionally subordinating market mechanisms to alternative, non-market logics, values, customs, and rules. This embedding strategy, of course, works to the extent that the non-market institutions within which markets are embedded embody values of justice, sustainability, efficiency, and pluralism. In traditional and tribal societies, embedding is ubiquitous; all markets are rooted in the institutions

that comprise the general fabric of social life. In market societies, this subordination of the market manifests in counter-movements to protect people and the rest of the web of life from its devastating encroachment [55,191]. This counter-movement can take several forms. I will describe each of these forms—reforms through the state, alternative markets, and non-market systems—and the barriers to achieving them. I argue that each of these counter-movements is an important but insufficient piece of efforts to align food systems with the normative objectives of ecological economics.

First, counter-movements can be reforms enacted through the state. These reforms change the rules of markets in ways that deviate from the self-regulating market system of economic theory, such as by constraining certain types of transactions or manipulating prices such that they are not entirely determined endogenously through supply and demand. This might include, for example, laws that limit or forbid speculating on agricultural commodities, to lessen the magnitude of food price shocks during times of shortage. Or it could consist of anti-hunger government programs like the Supplemental Nutrition Assistance Program (SNAP, or food stamps) in the United States, which provides limited-purpose money to low-income people with which they can purchase market food. Policies and programs that improve wages and working and living conditions for farmworkers or other food-system laborers also constitute a counter-movement against the disembedding of markets. Social movements for sustainable agriculture are part of the Polanyian counter-movement, too, to the extent that they push policies and food systems toward embodying non-market values [192].

Yet the realities of political economy limit the likelihood of achieving desirable food systems through such measures alone. Once markets exist, it is quite challenging to prevent indexes, derivatives, futures markets, and other speculative instruments from materializing [193], including illegally. Even the recent global financial crash did not lead to regulatory limits on finance. More troublingly, states use hunger for social control and as a rhetorical justification for their own interventions; thus, they do not want to fully eradicate its threat [163]. The SNAP program provides too little to afford a healthy diet [194,195], does not vary benefits with food prices, and exists as part of the U.S. farm bill, whose subsidies favor large-scale industrial agriculture [196] and reduce the price of foods whose consumption is associated with greater cardiometabolic health risks like obesity and high cholesterol [197]. Powerful corporate interests spend massive resources opposing effective regulations to protect labor and the environment [198]. Even well-intentioned policies often perpetuate injustice and unsustainability, especially in their effects beyond national borders. Any social and environmental protections achieved must be defended in perpetuity. Plus, changing rules and incentives constrains rather than transforms the fundamental logic of markets; it is hard to imagine how policy for market food systems could curtail instrumental values like selfishness or the drive to shift costs on others. The counter-movement to subordinate food markets to other social institutions and values through the state, like the drive to reduce inequality, is necessary but not sufficient and faces steep odds. Below, I elaborate further on the fundamental barriers to achieving desirable food systems through state action.

A second type of counter-movement involves constructing self-contained embedded markets, separate from the dominant commodity food system. These "alternative food networks" include farmers markets, consumer cooperatives, and direct sales from producers to institutions or local businesses [199,200]. They also encompass standards-based certification schemes like labels of origin [201], organic, and fair trade [202]. Alternative embedded markets can provide effective protections against certain undesirable features of food markets through formal and informal rules. Relational values related to justice and sustainability motivate many participants [203,204]. In alternative local markets, producers set prices with regard for more than market forces [205]. Local and urban food production tends to support justice and sustainability better than conventional agriculture for commodity markets [47].

Yet alternative food markets' contribution to creating more desirable food systems is complicated. Many scholars question whether the values and structures underlying alternative markets actually correspond with improved outcomes in terms of justice and sustainability [46,206–211]. Alternative-market producers must still prioritize financial viability and enact instrumental values,

after all [45]. Price premiums instrumentalize relational ethical values. Moreover, alternative markets are marginal: just 1.2 percent of the world's farmland is certified organic [212]; the global fair-trade market is one-tenth the size of organic [213]; and direct markets to consumers, institutions, or local businesses account for just over two percent of food sales in the United States [214]. Scaling up alternative food markets often means compromising their embeddedness in local social institutions or non-instrumental values [204,215]. Corporations have watered down certification standards and followed their regulations but not their principles, leading to contradictions like "industrial organic" farming [216–219]. On the other hand, alternative food networks provoke change in part through their relation to the dominant market food system, such as by pressuring major corporations to change their practices. Alternative food markets are one aspect of comprehensive movements, such as food sovereignty and agroecology, that seek to transform global food systems.

Protective policies and alternative markets make progress toward justice, sustainability, efficiency, and pluralism by constraining the market mechanisms' power over food systems. Another type of counter-movement is not regulating or embedding markets but creating non-market food systems. This can be done through the state or self-organization. I will treat each in turn. States, for their part, can centrally plan and organize food systems. In theory, they can govern all production and exchange of food above the level of the household without markets, as in the ideal of state communism, or they can administer small food systems or subsystems separately from the market, such as organizing food production and distribution for the military. In practice, neither of these examples tend to be fully non-market food systems according to my definition, since buying, selling, and prices are typically present. The Soviet Union and communist China, for example, both purchased the output of farms—whose operations were partly governed by the central planner—at prices set by the state, and then sold these products in state stores at another set of predetermined consumer prices [220]. Social programs to feed the poor or the military often involve the state purchasing food either at commodity prices or from contracted producers, and then either gifting that food or selling it at preset subsidized prices. Thus, actually existing state-run food systems are best characterized as markets that are highly embedded in authoritarian or bureaucratic social institutions. Whether by replacing markets or embedding them, state food systems can prioritize desirable values. Food rationing can contribute to justice by giving precedence to needs over wants, and to sustainability by limiting consumption [221,222]. State contracts that pay price premiums for agroecological farming can contribute to sustainability too [223]. Centrally planned economies, unlike those based on the private accumulation of capital, need not grow, at least in principle [224,225]. Centrally planned price schemes can make food markets stable, and also enable, by subsidy, the simultaneous realization of adequate remuneration for producers and affordable food for consumers.

But centrally organized state food systems have not performed desirably and may not be able to. China's horrific famine during the Great Leap Forward illustrates the worst possible injustices of state-planned food systems. From 1958 to 1961, 16–30 million people died prematurely—the greatest loss of human life to hunger in recorded history—mostly because of systemic failures in central planning: expecting implausible increases in productivity, China's government diverted resources from agriculture and procured too much food to send to cities, leaving farmworkers famished and unable to produce enough to feed their rural communities [226,227]. For achieving not just food security but also sustainability and efficiency, state bureaucracies' centralized knowledge is far less useful than the local ecological knowledge of food system participants spread across the landscape [63,228]. Productivist centrally planned economies in the Cold War years dedicated more resources to agriculture than market economies yet output remained less than desired [220]. In Cuba, the farms with the most autonomy over production decisions tend to fare better agroecologically—producing greater output from fewer inputs—than those subjected to central planning [224]. Moreover, the movement for food sovereignty is based on the premise that communities have the right to govern their own food systems [229]. Without mechanisms like the market that enable participants to express and respond to needs and

offers in a decentralized fashion, large-scale food exchange networks become woefully inefficient at allocating resources and nourishment to those who most need it.

There is reason to doubt that state-run food systems would ever be just even if planners were to have perfect information. Political elites tend to attend and respond to the desires of other elites, not ordinary people [230]. Those in positions of power leverage their status to personally benefit, consolidate their privilege, and extend it to those in their empathic circle [231]. They are hardwired to ignore risks that threaten less-privileged others or their own individualistic worldview, including their self-serving belief in meritocracy [232]. Regardless technical advances or political-economic system, elites eat first, even as marginalized people live on the edge of starvation [163]. States are history's only strict, fixed, extractive, bureaucratic social hierarchies [233]. When states produce or procure food to give away or sell at subsidized prices, even in market economies, it is the bread of "bread and circuses," provided to the hungry populace to quell unrest and manufacture consent. Food remains frequently used as a weapon of war in foreign policy [234,235]; it is an incredibly effective tool of coercion in a world of hungry humans. State grasps for power over domestic food systems should be seen in this context. What is more, markets are themselves a social institution like any other. Since states institute all markets, even market food systems are to a great extent planned [236]. Therefore, governments are largely responsible for the injustice, unsustainability, inefficiency, and value monism of real-world market food systems, too. States did not have to create market systems that approach the disembedded ideal of economic theory. But this is what they have tried to do, despite undesirable outcomes.

To be sure, state policies and programs *are* potentially effective means for working toward desirable food (and economic) systems, precisely because state governments have so much power. Historically, pressuring states to feed the hungry and generally improve social and environmental conditions has worked, especially when such pressure has organized itself in disobedient, leaderless mass movements with ambiguous demands [237–239]. Moreover, many policymakers, bureaucrats, and others in government truly do have good intentions; they care about justice, sustainability, efficiency, and honoring the plural values of constituents. What is remarkable is that *in spite of* declared noble intentions, hunger and poverty have never before been greater relative to the world's collective capacity to eradicate them [240]. They persist mainly because powerful elites have, deliberately or not, instituted economic systems that channel resources to themselves. International governance institutions like the United Nations continually tweak measurement methods and retroactively fabricate baselines to make it appear as if poverty and hunger are decreasing when they are not [9]. As ecological economists, devising policies to modify markets or programs to supplement them is an important part of our work; policymakers pick from available ideas. Striving for *better* markets and *better* states is striving for a better world. But what if there are actions that common people and marginalized communities can take to make food systems—and economies—more just, sustainable, efficient, and plural? What if they can avoid the pitfalls of markets and the state entirely? This is my argument for serious inquiry into the nature and potential of food systems without markets or states.

The call for regulated or embedded food markets misses grander opportunities for a more desirable world. Food is rarely traded in markets at all in contemporary and historical societies whose markets are embedded—that is, societies that do not subordinate social life and institutions to the market [42,58,241]. Internal (within-community) food markets arise when land and labor become commodities [55]. If, following Gerber and Gerber [22], ecological economics founds itself partly on freeing life from full determination by markets, we might do well to focus on freeing food—an essential, ecological, cultural good with unique characteristics that undermine many of the benefits of markets [242]. Think of food like health care. Economists have long struggled to reconcile market theory with the fact that general equilibrium cannot be reached if participants' survival is not guaranteed by some initial endowment [71]. A Nobel prize-winning economist, unable to find a satisfactory specification that did not assume death by starvation for those whose resources were insufficient, once conceded that the market model "would be found best suited for describing a society of self-sufficient farmers who do a little trading on the side" [243]—or any society whose nourishment is assured by

non-market food systems, I would add. In addition to alternative markets, food activists and scientists should consider alternatives *to* markets.

But the fact that markets mostly fail to produce justice, sustainability, efficiency, and value pluralism does not automatically entail that other economic arrangements for food systems can or will. Whether, and *how*, non-market food systems can succeed where markets fail is the key question around which to organize a research agenda on the topic for ecological economics. This is the final criterion of my rubric for assessing whether markets serve the public good, which I called "corruptness" in Table 1. Disembedded food markets fail the desirability criterion, I have argued, and embedded markets can approach desirability only partially and with great difficulty. Ecological economists must study how non-market food systems perform with respect to the normative foundations of our discipline. Figure 1 visually summarizes my argument.

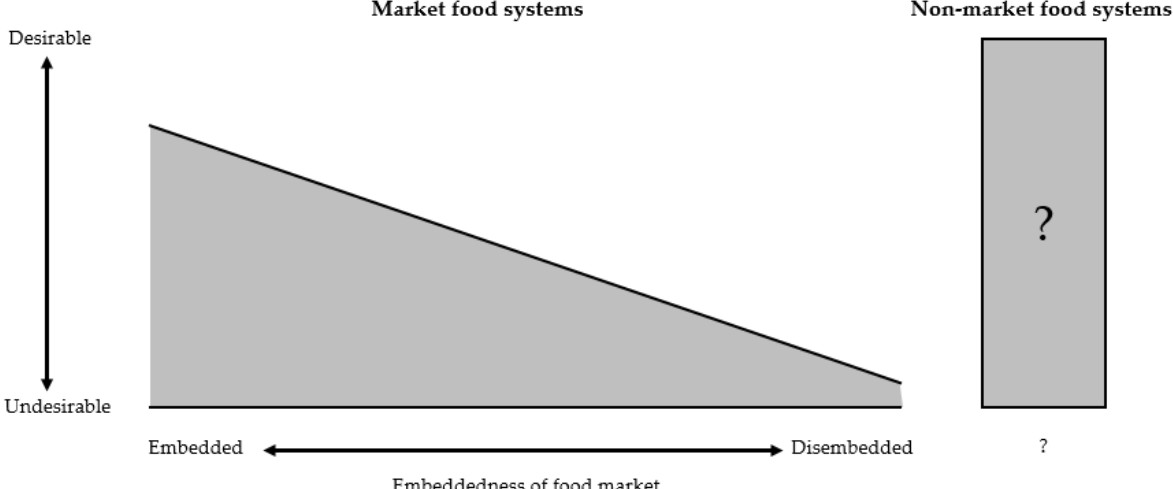

**Figure 1.** The gray areas represent possibility spaces for market and non-market food systems. Market food systems become increasingly homogenous and more necessarily "undesirable" (unjust, unsustainable, inefficient, value monist) as they are more disembedded from non-market social institutions. Non-market food systems can, in principle, be even more diverse than the most embedded markets, but they require further study to assess their desirability.

Of course, directly comparing market and non-market food systems, or different types of non-market food systems with each other, is tricky because all else is never equal. Yet learning about how non-market food systems function can point the way toward an understanding of their role in transformation toward more desirable food regimes. Through empirical analysis, ecological economists can determine what sorts of non-market food systems to promote based on their performance or potential with relation to justice, sustainability, efficiency, pluralism, or the values that participants themselves hold. Theoretical work can then contemplate how such systems might replicate themselves or come together in desirable assemblages of coexisting food systems. A subsequent article will review research on non-market food systems and suggest an agenda for ecological economists studying them. My purpose is not to propose a blueprint for desirable non-market food systems, but to suggest that ecological economists examine those that already exist in every society on earth. Moreover, since non-market food systems are created spontaneously and autonomously by communities, not planned or instituted by governments, it is unclear to whom a scholar would propose how to create a desirable one. Here, I will conclude by reviewing the broad outlines of the research agendas I have proposed for critically assessing the desirability of food system institutions and plans to transform them.

## 7. Conclusions

Overall, today's global food systems are unjust, unsustainable, inefficient, and value-monist. Yet the primary institution for governing them, markets, has hardly been questioned as such. I have argued that markets bear much responsibility for the undesirable nature of food systems. In so doing, I have proposed a rubric of sorts for assessing the ecological–economic desirability of markets for food, and I put forth several contentions and hypotheses intended to initiate research and incite debate around this question. Researchers can and should adapt this agenda for considering consistency with the normative foundations of ecological economics to any economic institution, not just markets, and any good or service. I focused on markets because of their ubiquity and acceptance, and on food because of its status as an essential, ecological, and culturally important resource.

To evaluate the desirability of markets as such, I argue that one should pay attention to markets that are disembedded from other social institutions. To sum up, markets allocate food to its most lucrative uses, not the hungriest humans. People act selfishly and accept injustice in market settings. Market pressures force food producers to shift costs onto the public and ecosystems. Market prices rarely signal environmental degradation. Market competition in food systems drives the economic growth that has pushed resource use and waste generation past planetary thresholds of sustainability. Markets for food are unstable and unlike the efficient markets of economic theory. They revolve around monetary value, neglecting food systems' cultural, spiritual, and ecological attributes.

Despite our compelling economy-in-society-in-nature diagrams, ecological economists' most typical methods are well suited for studying economic systems as separate spheres, divorced from but interacting with their social, cultural, political, and biophysical milieus. But to study embedded markets and non-market systems, where no separate economic institutions exist, one must understand the economy as just one aspect of an integrated whole made of nature, culture, social organizations, and supernatural meta-persons. This requires developing what Clovis Cavalcanti has tentatively called ethnoecological economics [244]. This transdisciplinary literature review, like the synthesizing work of social ecological economics [67], coevolutionary ecological economics [245], or political ecological economics [121], works toward fulfilling our field's holistic intentions. With a broader set of quantitative and qualitative tools, as well as more diverse theoretical frameworks to draw on, researchers are better equipped to critically consider the feasibility and desirability of different options for embedding food markets through policy or alternative food networks, or for maintaining and creating non-market food systems—an astoundingly underdeveloped area of inquiry.

Thinking about how the evolution of economic institutions interacts with justice, sustainability, efficiency, and values will not end with a convincing set of answers. Nor is it meant to. This research agenda's purpose is to deepen and sharpen our understandings of the ways in which communities work toward and at times achieve these goals (or not), in service of transforming societies toward them. This research is meant to inform action. If my argument holds any kernel of truth, if market food systems are undesirable and all strategies for resolving their shortcomings are partial and extremely challenging, then this in itself warrants substantial promotion and propagation of non-market food systems. Research is part of action; ecological economists should also analyze, experiment with, and theorize about non-market food systems. We should learn from those who produce food that is not for sale and exchange food without money. We should assess diverse non-market food systems' desirability according to the rubric presented in this article.

Research is action in a more fundamental sense, too. Researchers create reality as they study it [246]. Data is generated, described, modified, analyzed, and interpreted; it is not simply out there waiting to be discovered or harvested. By drawing academic attention to non-market food systems, researchers bring them into being in the minds of their participants and give them legitimacy in society. I cannot explain my research to an interviewee without familiarizing them with the critiques of food markets or the concept of non-market food systems. Research is political, not only in its philosophical orientation but in the subjects we decide to study [247]. Theories of food systems, similarly, not only reflect reality but shape it [60]. Because social facts and values are inseparable, this article unavoidably

criticizes not just market-based food systems but also the *idea* that markets are compatible with desirable food systems. May the ensuing debate bear fruit.

**Funding:** This research was funded by the Social Sciences and Humanities Research Council of Canada (SSHRC), grant number 895-2013-1010.

**Acknowledgments:** Thanks to all my colleagues at the University of Vermont and in the Economics for the Anthropocene initiative who have questioned my ideas while encouraging this line of inquiry. I am especially appreciative of Bengi Akbulut and Dan Tobin for encouraging me to adopt a Polanyian framework for thinking about markets, of Dan for asking me what a market is, and of Josh Farley for preparing the fertile ground for this research project.

**Conflicts of Interest:** The author declares no conflict of interest.

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
