# Peer review of "The Case for Studying Non-Market Food Systems"

_sustainability, doi:10.3390/su11113224_

Reviewer 1 Report

The revision of The case for studying non - market food systems has been challenging. I found the paper really interesting, the approach accurate and the perspective innovative. I have just some minor doubts. First of all, why title? I think it is not meaningful. Moreover, I do not like the tone and the structure of abstract. To help the reader, I think that the methodological approach has to be more widely explain, in particular the paper selection. 

Author Response

(R1) The revision of The case for studying non - market food systems has been challenging. I found the paper really interesting, the approach accurate and the perspective innovative. I have just some minor doubts.

Thank you for engaging with, and finding value in, a paper you have found challenging to review.

(R1) First of all, why title? I think it is not meaningful.

I appreciate the focus on choosing a good title, as this decision to a large extent determines the reach and impact of an academic paper. I disagree, however, with the opinion that my title is not meaningful. This article is a contribution to a special issue on a “research agenda for ecological economics” and the title succinctly sums up my main contribution to that research agenda: an argument for focusing research attention on non-market food systems. Of course I could have added “as well as non-market production and distribution systems for other essential, ecological, and/or culturally important resources” but this would take away the title’s clarity, brevity, and thus its power, in my view.

A previous iteration of this essay had the title “Markets inhibit progress toward justice, sustainability, efficiency, and value pluralism in food systems,” which reflects the argument to which I have dedicated the most space in this argument. I could have modified that to something like: “Market governance of food systems is barrier to achieving the normative goals of ecological economics.” But I think that this sort of title risks overstating my argument by stripping all the nuance that the article’s Polanyian approach of embeddedness and disembeddedness brings. Perhaps I could have titled it, “The more disembedded the food market, the less desirable the food system” or “Increasingly disembedded market food systems increasingly inhibit progress toward justice, sustainability, efficiency, and value pluralism.” However – and this is the reason I am keeping the original title I submitted – stating my argument or finding in the title, rather than my contribution to the ecological economics research agenda, makes it unclear why the article belongs in the special issue.

My argument is not that food markets are bad as such and I do not want to suggest this in the title. My argument is that there are reasons to be curious about how some types of non-market food systems might better realize the normative foundations of ecological economics, and ecological economists should study this possibility.

(R1) Moreover, I do not like the tone and the structure of abstract. To help the reader, I think that the methodological approach has to be more widely explain, in particular the paper selection.

I am not able to respond to the general dislike of the abstract’s tone and structure since no specific criticisms or recommendations were given. The call for including more information about the methodological approach, though, is a good one. My original abstract said what I argued, but not how I went about making this argument. To signal that the article is a transdisciplinary literature review I added the clause “Drawing on empirical and theoretical literature from diverse intellectual traditions,” to the beginning of the sentence that before stated only, “I argue that markets, as an institution for governing food systems, hinder the realization of these objectives.”

I could not tell if the comment about better explaining paper selection referred only to the treatment of methods in the abstract, or in the article’s “methods” section as well. In the methods section, I do not detail how I found every book and article I cite in the ensuing results section; instead I explain and defend my choice of disciplines whose work I draw on to make my argument. This is common practice in a literature review – see contributions to the Annual Review of … journals, for example. Only in systematic literature reviews must the author describe and justify their search keywords and their criteria for inclusion and exclusion of studies. The main text of this article is not a systematic literature review. (I did, however, perform a systematic literature search of studies on non-market food systems published in the journal Ecological Economics, whose results and methods I describe in the lengthy footnote on the article’s second page.)

Reviewer 2 Report

Even if I don't fully agree with the authors' point of view, I think that manuscript is well written and authors' opinion are well expressed. So I think that manuscript should be considered for publication by the editor to open a debate on the issue. My personal point of view is that we are not facing a food market in the world but several markets, having a different spatial dimension (from local to global), selling highly differentiated food (from commodities to highly valued traditional food) to different consumers topologies (no-low vs high income, urban population vs rural one, etc). Consequently tailored approaches to each specific context have to be addressed through adequate social and market policies to correct/limit market failures. However I guess that marketS approaches are inevitable.

Author Response

(R2) Even if I don't fully agree with the authors' point of view, I think that manuscript is well written and authors' opinion are well expressed. So I think that manuscript should be considered for publication by the editor to open a debate on the issue.

Thank you for recognizing the value of an intervention you do not fully agree with. This suggests I have made my arguments rigorously. I respond to your comments below but have not made any specific changes because none were suggested. That said, I appreciate and largely agree with the “personal point of view” you share.

(R2) My personal point of view is that we are not facing a food market in the world but several markets, having a different spatial dimension (from local to global), selling highly differentiated food (from commodities to highly valued traditional food) to different consumers topologies (no-low vs high income, urban population vs rural one, etc).

I agree with this perspective, and I hope that a close reading of the article reflects my agreement. The paragraphs that begin with “There are many types of markets” lay out my understanding of the diversity of markets. This is why I use “markets” not “market” throughout the article. The global commodity markets are, in my Polanyian framework, the most “disembedded” possible food markets, whereas local, traditional food markets tend to be “embedded” in social institutions other than markets.

(R2) Consequently tailored approaches to each specific context have to be addressed through adequate social and market policies to correct/limit market failures.

From this article’s Polanyian perspective, policy is a way to further embed markets in social institutions (of the state, in most cases) and this will support food systems’ realization of the normative foundations of ecological economics – sustainability, justice, efficiency, value pluralism – to the extent that the social institutions within which a given market is embedded embody those objectives. I recognize that this may be an overgeneralization, or at least an oversimplification of the complex problems involved in designing good food policy. My argument, though, only points out the general limitations of policy remedies to the problems posed by markets as an institution for governing food systems; it does not evaluate or propose specific policies, which must be, as you say, context specific.

(R2) However I guess that marketS approaches are inevitable.

This comment corresponds with Section 3, titled “food markets seem inevitable,” in which I lay out the reasons that scholars tend to believe that food markets are inevitable. I do not think food markets are inevitable. Many societies – and every prehistoric society, evidence suggests – have no food markets at all. But neither am I advocating the abolition of food markets. Non-market food systems exist in every society on earth, alongside markets where markets exist. So, even if markets for food are inevitable beyond a certain size and complexity of a society, this in no way refutes any piece of my argument.

Reviewer 3 Report

The paper presents very interesting research which fits to the scope of the journal. This review of different aspects of current global food systems contain highly valuable and necessary critical discussion on the purpose and future directions in food systems management. However, I miss some balancing some information by showing both sites of the discussed issue. I present some comments which should be considered in order to improve the quality of the paper.

In the introduction there is a reference to the use of food for biofuel production “Meanwhile, 9 percent of global crop calories feed biofuel refineries and other industrial processes instead of humans.” However, the reference [10] Cassidy, E.S.; West, P.C.; Gerber, J.S.; Foley, J.A. Redefining agricultural yields: from tonnes to people nourished per hectare. Environ. Res. Lett. 2013, 8, 034015 presents different statement “Together crops used for industrial uses, including biofuels, make up 9% of crops by mass, 9% by calorie content (…)”. Therefore, assigning whole share only to biofuels is not adequate. Moreover, I suggest to discuss one more issue in the context of biofuel production. There are different generations of biofuels (four generations). While the first generation biofuels rely on food crops as their feedstock, the second generation biofuels use other components, usually lignocellulosic biomass that cannot be used as a food for humans. Of course, there is a problem of using the agricultural land for dedicated biofuel crops which intentionally are not concentrated on the food production, but on the growth of biomass for energy production. Nevertheless, in food-oriented production there are wastes that can be used for biofuel production without any harm to human nutrition.

I miss an important discussion on the local food production which is strongly connected with non-market food systems. Based on the problem of depleting resources in the countries of the global south and “food miles” there are some local actions undertaken to manage local food systems, for instance: ecological food footprint as a measure of environmental carrying capacity, foodshed of urban areas that is necessary to guarantee local food production, or urban agriculture as a nature-based solution. All of these approaches have an impact on growing systems of community supported agriculture, sometimes based on barter transactions. See for example:

· Application of Ecological Footprint Accounting as a Part of an Integrated Assessment of Environmental Carrying Capacity: A Case Study of the Footprint of Food of a Large City. Resources 2018, 7, 52.

· Foodshed as an Example of Preliminary Research for Conducting Environmental Carrying Capacity Analysis. Sustainability 2018, 10, 882.

· The Role of Urban Agriculture as a Nature-Based Solution: A Review for Developing a Systemic Assessment Framework. Sustainability 2018, 10, 1937.

As these are emerging topics many other papers in that area can be found. In my opinion including these elements could improve the paper by showing both sites and completing the content of section 5.2. Food markets are unsustainable by presenting relatively rare but still the actions that are undertaken in the current food system.

I hope that my suggestions would be helpful to improve the paper. It was really a pleasure to review this article.

Author Response

(R3) The paper presents very interesting research which fits to the scope of the journal. This review of different aspects of current global food systems contain highly valuable and necessary critical discussion on the purpose and future directions in food systems management.

Thank you.

(R3) However, I miss some balancing some information by showing both sites of the discussed issue. I present some comments which should be considered in order to improve the quality of the paper.

In Section 4, Methods, lines 224-227, I write “My argument is decidedly one-sided. For the reader interested in reviewing the theoretical advantages and well-rehearsed defenses of markets, any elementary economics textbook will do. My purpose is to synthesize theory and evidence from diverse disciplines to call food markets’ desirability into question.” Mainstream economics has vigorously defended the desirability of markets. In ecological economics and related disciplines, it has been typical to question the desirability of markets (such as the literature I draw on for this article’s section 5. Results), yet then assume that markets are inevitable (for the reasons I enumerate in section 3. Food markets seem inevitable). My goal in this paper is to systematically call into question the desirability of markets as an institution for governing food systems, according to the normative foundations of ecological economics, by drawing on existing literature.

That said, I appreciate your suggestions for making my argument clearer and more inclusive of contrasting positions.

(R3) In the introduction there is a reference to the use of food for biofuel production “Meanwhile, 9 percent of global crop calories feed biofuel refineries and other industrial processes instead of humans.” However, the reference [10] Cassidy, E.S.; West, P.C.; Gerber, J.S.; Foley, J.A. Redefining agricultural yields: from tonnes to people nourished per hectare. Environ. Res. Lett. 2013, 8, 034015 presents different statement “Together crops used for industrial uses, including biofuels, make up 9% of crops by mass, 9% by calorie content (…)”. Therefore, assigning whole share only to biofuels is not adequate.

I do not assign the “whole share only to biofuels.” You quote me: “9 percent of global crop calories feed biofuel refineries and other industrial processes,” and quote Cassidy et al.: “crops used for industrial uses, including biofuels, make up 9% …” We are saying the same thing.

(R3) Moreover, I suggest to discuss one more issue in the context of biofuel production. There are different generations of biofuels (four generations). While the first generation biofuels rely on food crops as their feedstock, the second generation biofuels use other components, usually lignocellulosic biomass that cannot be used as a food for humans. Of course, there is a problem of using the agricultural land for dedicated biofuel crops which intentionally are not concentrated on the food production, but on the growth of biomass for energy production. Nevertheless, in food-oriented production there are wastes that can be used for biofuel production without any harm to human nutrition.

Thank you for adding some nuance to my discussion of biofuels. However, I only mention three times in the article. I never critique biofuels as such, only market allocation of edible crops (or the produce of cropland) to biofuel production. For this reason, I incorporate the suggestion as a footnote to line 290, after “Many people cannot afford enough market food to meet their basic nutritional needs. Meanwhile, others pay to overeat, waste food, and direct edible crops to livestock and biofuel production.6” This footnote [6] reads: “Note that biofuels can be produced from non-food crops. In this case, the use of agricultural land for energy crops instead of food crops can still have perverse consequences for justice and sustainability. Nevertheless, food production often produces wastes that are useful as biofuel feedstock and not very useful for human nutrition.”

(R3) I miss an important discussion on the local food production which is strongly connected with non-market food systems. Based on the problem of depleting resources in the countries of the global south and “food miles” there are some local actions undertaken to manage local food systems, for instance: ecological food footprint as a measure of environmental carrying capacity, foodshed of urban areas that is necessary to guarantee local food production, or urban agriculture as a nature-based solution. All of these approaches have an impact on growing systems of community supported agriculture, sometimes based on barter transactions. See for example:

· Application of Ecological Footprint Accounting as a Part of an Integrated Assessment of Environmental Carrying Capacity: A Case Study of the Footprint of Food of a Large City. Resources 2018, 7, 52.

· Foodshed as an Example of Preliminary Research for Conducting Environmental Carrying Capacity Analysis. Sustainability 2018, 10, 882.

· The Role of Urban Agriculture as a Nature-Based Solution: A Review for Developing a Systemic Assessment Framework. Sustainability 2018, 10, 1937.

As these are emerging topics many other papers in that area can be found. In my opinion including these elements could improve the paper by showing both sites and completing the content of section 5.2. Food markets are unsustainable by presenting relatively rare but still the actions that are undertaken in the current food system.

Yes, the topics of local and urban food production are highly relevant to studying both non-market food systems and embedded food markets. Here’s my response to your comments and suggestions:

Urban agriculture and local markets tend to be highly embedded markets, as I imply in the last paragraph of section 2 (lines 144-146) by adding the words “embedded” and “disembedded” to make the sentence clearer: “local, weekly embedded marketplaces each have their own norms and quirks, while disembedded markets for agricultural commodities like wheat are global and rather homogenous.” Section 5.2. (and all of section 5. Results) refers to tendencies of disembedded markets. Therefore information about the sustainability benefits of urban agriculture and local food would be inappropriate to include in section 5.

But, I did add a sentence citing the work on ecological footprint accounting of a city’s food purchases in section 5.2: “City dwellers purchase food whose production may be invisibly unsustainable.”

In section 6, I refer to local food as an example of embedding markets by creating “alternative food networks” that can achieve the normative goals of ecological economics better than disembedded markets. I added a sentence based on your suggestions and sources: “Local and urban food production often uses fewer resources and generates less waste than conventional agriculture for commodity markets.”

Finally, you suggest, “local food production … is strongly connected with non-market food systems. … sometimes based on barter transactions.” This is an excellent point. I added a sentence in the introduction (lines 62-65): “While the emerging body of research on urban farming and local food has focused on commercial production and exchange [45–47], food systems based on reciprocity, redistribution, and self-production are nearly always local.”

(R3) I hope that my suggestions would be helpful to improve the paper. It was really a pleasure to review this article.

Yes, this was helpful. I am grateful.

Reviewer 4 Report

In abstract I would expect at least some word on methodological approach, what will be studied etc. In present form, abstract provides information only about problems and conclusions. After reading a whole paper I know that this is a kind of critical literature review. However, it should be stated in a clearer way.

The main advantage of the article is a careful review of literature in the context of the functioning of modern markets, especially the food market. The review is quite one-sided in nature, but the author points out in advance, which position he will represent. The arguments he raised are usually not new but the compilation of these arguments in one paper is valuable to the reader.

 It is difficult to refer to particular arguments in favour of criticism of contemporary food markets because of the multitude of issues raised.  However, due to the large volume of the article and the lack of any materials in tabular form, the article is quite difficult to read. Its reception would definitely improve the presentation of the basic axis of argumentation (along with key references), e.g. in the form of tables or graphs.

However, the author should point out more clearly that the described weaknesses concern not only the food market, but also many other markets. Moreover, it should be more clearly indicated that the problem concerns a specific model of functioning of agriculture and the food market. This model can be assessed as industrial. In the paradigm of sustainable agriculture, many of the described problems are, at least partially, solved. In this context, the article lacks, first of all, a reference to the common agricultural policy implemented in the countries of the European Union. The justification of this policy results from, among other things, the need to provide 4 elements (justice, sustainability, efficiency, value pluralism) which the author made the axis of his analysis.

The main contribution of the author is to point to the needs of research on alternative food systems . However, the idea put forward by the author should be extended and clarified. At the moment, it is clear that the author hopes for a non-market system, but it would be worthwhile to propose in a more accessible form the main principles for the functioning of this system.

Author Response

(Reviewer 4) In abstract I would expect at least some word on methodological approach, what will be studied etc. In present form, abstract provides information only about problems and conclusions. After reading a whole paper I know that this is a kind of critical literature review. However, it should be stated in a clearer way.

Thank you for this suggestion. One other reviewer also asked me to provide more information about the methodological approach in the abstract. My original abstract said what I argued, but not how I went about making this argument. To signal that the article is a transdisciplinary literature review I added the clause “Drawing on empirical and theoretical literature from diverse intellectual traditions,” to the beginning of the sentence that before stated only, “I argue that markets, as an institution for governing food systems, hinder the realization of these objectives” [lines 11-13].

(R4) The main advantage of the article is a careful review of literature in the context of the functioning of modern markets, especially the food market. The review is quite one-sided in nature, but the author points out in advance, which position he will represent. The arguments he raised are usually not new but the compilation of these arguments in one paper is valuable to the reader.

Thanks.

(R4) It is difficult to refer to particular arguments in favour of criticism of contemporary food markets because of the multitude of issues raised.  However, due to the large volume of the article and the lack of any materials in tabular form, the article is quite difficult to read. Its reception would definitely improve the presentation of the basic axis of argumentation (along with key references), e.g. in the form of tables or graphs.

I can imagine 3 possible visuals for this article. This first two I have made and put into the text, and I thank you for providing the impetus to include them. The third, which may have been the one you were calling for, I have decided not to create, for reasons I explain below.

1. A table showing the criteria for determining whether markets serve the public good in the case of any specific good, service, or resource. I have added this table to the article [Table 1]

Criterion

Markets should govern the production and exchange of a good   if…

Rivalry

its   consumption subtracts from it or prevents others from consuming it

Excludability

institutions   can prevent specific actors from accessing or consuming it

Non-fictitiousness

it   is produced for sale

Complexity

it   is produced and exchanged in complex networks of actors

Desirability

markets   can promote justice, sustainability, efficiency, and value pluralism

Corruptness

non-market   institutions would do so undesirably

2. A figure that shows, abstractly, how markets become increasingly homogenous and more necessarily “undesirable” (unjust, unsustainable, inefficient, value monist) as they are more disembedded from non-market social institutions, and how non-market economies may be even more diverse than “fully” embedded markets.  I have inserted this diagram at line 888.

3. A table that summarizes my arguments regarding disembedded market food systems’ injustice, unsustainability, inefficiency, and value monism, with references. This table, while it could be useful to the reader, runs the risk of oversimplifying complex arguments. I have chosen not to present the information in the form of a table or graph because my arguments are not the results of a systematic literature review. I fear falling into the “fallacy of misplaced concreteness” by presenting my findings as if they were the only or best possible interpretations of a set of evidence. That said, I appreciate that it might make it easier for the reader to navigate the article with some visual simplifications of the arguments.

(R4) However, the author should point out more clearly that the described weaknesses concern not only the food market, but also many other markets.

I address this in a paragraph in section 2. Lines 136-143:

“Markets for foods are in many ways their own type [60]. Food is a human physiological necessity, without which we do not exist. Foods are organisms that comprise the ecosystems of which we form part. Food is the basis of rituals in every culture. Markets for other things that are essential, ecological, or culturally important share some of the characteristics of food markets I describe below. Some aspects of food markets apply to virtually all markets. Therefore, many of the hypotheses and contentions I make below can inform a research agenda questioning whether markets serve the public good in the case of not just food but any chosen good or service, especially other essential, ecological, and culturally important resources such as housing, water, or medicines.

At the end of section 7, too, I write in lines 921-923: “Researchers can and should adapt this agenda for considering consistency with the normative foundations of ecological economics to any economic institution, not just markets, and any good or service.”

(R4) Moreover, it should be more clearly indicated that the problem concerns a specific model of functioning of agriculture and the food market. This model can be assessed as industrial.

Yes. Because this article is about the economic systems within which food systems exist, I call this model the “disembedded” market food system. That it has co-evolved with industrial agricultural practices, though, is a good point. I make this point in lines 305-311, where I argue that market mechanisms have facilitated the increase in scale of production and size of firms in the food system. I added a sentence in lines 406-407 that says so explicitly, in the section on sustainability: “Over time, market mechanisms have facilitated the industrialization of the food system, which has wrought havoc on ecosystems around the world.”

(R4) In the paradigm of sustainable agriculture, many of the described problems are, at least partially, solved.

By the same token, sustainable agriculture has co-evolved with “embedded” market food systems – organic farming, local markets, etc. Section 6 describes how embedded markets partially resolve many of the described problems of disembedded food markets. Sustainable agriculture can be practiced in embedded markets or in non-market food systems.

(R4) In this context, the article lacks, first of all, a reference to the common agricultural policy implemented in the countries of the European Union. The justification of this policy results from, among other things, the need to provide 4 elements (justice, sustainability, efficiency, value pluralism) which the author made the axis of his analysis.

The justification for the common agricultural policy does stem from justice, sustainability, efficiency, and some values. This makes it a case of embedding food markets through policy in pursuit of desirability, which I describe in lines 736-765. It is not clear why this case would be worth mentioning as an example, though. In fact, many have criticized the EU common agricultural policy for creating outcomes that are unjust (most subsidies go to large farms and cheap food gets dumped on the global south, undercutting peasant farms who cannot compete with subsidized industrial agriculture), unsustainable (guaranteed prices make it economically worthwhile for farmers to use all available land, with chemicals, to grow more crops than are demanded), inefficient (subsidies distort the market), and value monist (the EU policy making process prioritizes economic values over others). See Jeffery, Simon. 2003. “The EU Common Agricultural Policy.” The Guardian, June 26, 2003, sec. World news. https://www.theguardian.com/world/2003/jun/26/eu.politics1.

In fact, this might be a better example of the limits of embedding markets in government institutions through policy.

(R4) The main contribution of the author is to point to the needs of research on alternative food systems . However, the idea put forward by the author should be extended and clarified. At the moment, it is clear that the author hopes for a non-market system, but it would be worthwhile to propose in a more accessible form the main principles for the functioning of this system.

I am suggesting that ecological economists study the non-market food systems that already exist in every society on earth. Perhaps integrated proposals for desirable food systems of the future will arise from such a research agenda, but my point is that as yet not enough is known to prescribe anything specific. Moreover, since non-market food systems are created spontaneously and autonomously by communities, not planned or instituted by governments, it is unclear to whom a scholar would propose how to create a desirable one. Imagining might be worth it, though. I have added some sentences to this effect in section 6, line 908-912

Round  2

Reviewer 1 Report

I did my indications. The author managed accordingly to his view. 

Author Response

Thanks again for your thoughtful comments.

Reviewer 4 Report

None of my comments have been addressed. 

Author Response

I did not receive your comments with the first round of reviews. I have addressed them now, in a reply to your first review report. Thank you for taking the time to constructively critique the paper. It has improved as a result.

Round  3

Reviewer 4 Report

I accept the authors’ comments to my review.